# Unveiling the latent dynamics in social cognition with multi-agent inverse reinforcement learning

## Abstract

Understanding the intentions and beliefs of others, a phenomenon known as "theory of mind", is a crucial element in social behavior. These beliefs and perceptions are inherently subjective and latent, making them often unobservable for investigation. Social interactions further complicate the matter, as multiple agents can engage in recursive reasoning about each other's strategies with increasing levels of cognitive hierarchy. While previous research has shown promise in understanding a single agent's belief of values through inverse reinforcement learning, extending this to model multiple agents remains an open challenge due to the computational complexity. In this work, **we adopted a probabilistic recursive modeling of cognitive levels and joint value decomposition to achieve efficient multi-agent inverse reinforcement learning (MAIRL)**. We provided a numerical method to evaluate value decomposition errors in multi-agent tasks with discrete state and action spaces. To validate our method, we conducted simulations of a two-agent cooperative foraging task in a grid environment. Our algorithm revealed the ground truth goal-directed value function and effectively distinguished between level-0 and level-1 agents. When applied to human behavior in a cooperative hallway task, our method identified meaningful goal maps that evolved with task proficiency and an interaction map that is related to key states in the task without accessing to the task rules. Similarly, in a non-cooperative task performed by monkeys, we identified mutual predictions that correlated with the animals' social hierarchy, highlighting the behavioral relevance of the latent beliefs we uncovered. Together, our findings demonstrate that MAIRL offers a new framework for uncovering human or animal beliefs in social behavior, thereby illuminating previously opaque aspects of social cognition.

## 1 Introduction

Humans are remarkable at understanding the mental states of others - their beliefs, values, and intentions - a capacity known as "theory of mind (ToM)." At its core, an individual's behavior is often not directly guided by objective reality, but by their subjective beliefs about the state of the world. "Theory of mind" thus allows us to more reliably predict other people's actions instead of using the physical reality or one's own belief (Frith & Frith, 2012). This reasoning process can operate recursively, with individuals modeling not just others' beliefs, but others' beliefs about their own beliefs - a phenomenon captured by increasing levels of cognitive hierarchy. Precisely inferring these often unobservable beliefs and subjective values is crucial for understanding social cognition in humans as well as other species. We adopt a multi-agent inverse reinforcement learning (MAIRL) framework to achieve such inference. Addressing a crucial challenge in neuroscience and cognitive science, our novel approach promises to reveal computation of value and mental states of multiple interacting agents in social behavior.

Multi-agent reinforcement learning (MARL) models interactions among multiple agents (Yang, 2021)(Buşoniu et al., 2010). Despite its wide application in topics including traffic coordination (Campos-Rodriguez et al., 2017), wireless sensor networks (Derakhshan & Yousefi, 2019), and patient coordination in healthcare (Shakshuki & Reid, 2015), its application in neuroscience and cognitive sciences has been limited. There are many categories of MARL algorithms but we focus

mainly on those with centralized training and different levels of execution, in which agents share a common critic, but execute policies independently. A review of relevant algorithms can be found in this benchmark paper (Papoudakis et al., 2020).

In many social interactions, an agent anticipates the behaviors of others by constructing predictive models of them. This essential aspect of interaction is captured by recursive probabilistic reasoning, often used in the field of opponent modeling (OM) (De Weerd et al., 2013)(Albrecht & Stone, 2018)(Tejwani et al., 2022). However, OM is often not incorporated into modern RL frameworks due to computational expense and the need for centralized training to allow access to the ground truth strategies by all agents (Wen et al., 2019). Nevertheless, in well-defined social tasks, participants are expected to fully understand the rules, objectives, and their opponents' strategies. This understanding justifies the use of recursive modeling and the assumption that participants will make the best responses based on their predictions. Additionally, recent approaches have aimed to decouple agents' mutual effects during training through variational Bayes methods, enabling the approximation of task solutions (Wen et al., 2019) and opening more possibilities for large-scale OM.

While multi-agent reinforcement learning (MARL) with recursive reasoning typically models social behavior in goal-directed tasks, real-world interactions among humans or animals often occur without explicitly defined value functions. Inverse reinforcement learning (IRL) addresses this by inferring the latent value functions from observed behaviors (Arora & Doshi, 2021), thereby uncovering an individual's model of the world. Assuming the value function is a linear combination of known feature functions, the problem can be formulated as a linear optimization to maximize expert feature expectations, with various approaches addressing computation complexity and solution ambiguities (Abbeel & Ng, 2004)(Ratliff et al., 2006)(Ziebart et al., 2008). We focused on the maximum entropy approach proposed in (Ziebart et al., 2008), as it provides a probabilistic version of the problem, facilitating the incorporation of recursive reasoning crucial for ToM modeling.

Merging these lines of work, our approach uncovers both goal related value functions of the agents and mental models of one agent for another. In Section 2, we first review related MAIRL theoretical work in the field of imitation learning, and existing work that model social interactions in neuroscience and cognitive sciences. In Section 3, we describe our formulation based on multi-agent Markov Decision Processes (MDP) with recursive reasoning and value decomposition to model two expert agents in simultaneous-move games and a maximum a posteriori algorithm for inference. We proved that the approximation error of adopting value decomposition is negligible through mathematical analysis and numerical evaluation. In Section 4, we show that our model **effectively recovers goal-related value functions, interaction terms associated with task rules, and the levels of cognitive hierarchy between two agents** in a simulated multi-state, multi-action cooperative foraging task performed in a grid environment. We further apply our model to experimental datasets in humans and non-human primates, uncovering previously unobservable value maps and latent strategies used by the players.

## 2    RELATED WORK

Multi-agent inverse reinforcement learning (MAIRL) has been extensively studied in imitation learning, primarily aiming to enhance task performance. In cognitive science, models of social interactions often rely on single-agent Partially Observable Markov Decision Process (POMDP) or multi-agent Markov Decision Process (MDP), with simplified mentalization processes parameterized by only one or two variables. This work proposes adopting MAIRL with value decomposition and recursive reasoning for applications in cognitive sciences and neuroscience to enable efficient and interpretable value function recovery.

MAIRL studies in imitation learning have established a robust theoretical framework and practical tools for broader applications. The primary objective is to derive compactly parameterized reward functions that *rationalize* observed behaviors or equilibria. However, the problem is often ill-posed due to the vast solution space, necessitating constraints for tractability. For instance, Natarajan et al. (2010) aimed to maximize the difference between the value of group-optimized actions and alternatives to enable centralized control for population benefits. Similarly, Waugh et al. (2013) constrained learned value functions to ensure no greater regret than observed behaviors, while Reddy et al. (2012) used Nash equilibrium to decompose joint rewards into individual ones. Wu et al. (2023) decentralized reward estimation by inferring policy distributions of other agents. On the theoretical front, advancements in scalability are particularly promising. Yu et al. (2019) proposed multi-agent

adversarial inverse reinforcement learning to solve MAIRL in high-dimensional continuous state-action spaces under a logistic quantal response equilibrium. Kuleshov & Schrijvers (2015) proved that MAIRL in succinct games can be solved using optimization algorithms with polynomial complexity in the number of players, contrasting with traditional methods that are exponential.

In cognitive and neuroscience research, models of social interactions often focus on Bayesian inference over counterparts' strategies to maximize task outcomes. Khalvati et al. (2016) modeled social games with binary decisions as single-agent POMDPs, treating group conformity as a hidden state, which is fully characterized by two parameters. (Wu et al., 2021) and Ullman et al. (2009) employed multi-agent MDPs to infer binary intentions and decentralize policy generation for improved or more human-like task performance. Yoshida et al. (2008) used multi-agent MDPs to model recursive Theory of Mind (ToM), assuming distinct value functions for different cognitive hierarchies. Another line of modeling approaches seeks to uncover latent factors in animal behavior, such as goals (Baker et al., 2009) or context-specific intentions (Velez-Ginorio et al., 2017). Baker et al. (2017) used single-agent POMDPs to comprehensively predict human inference by modeling beliefs, desires, and percepts. (Shuvaev et al., 2024) analyzed social conflict paradigms with multi-agent POMDPs, parameterizing policies with Bernoulli processes to infer aggression levels.

## 3 MAIRL: MULTI-AGENT INVERSE REINFORCEMENT LEARNING

### 3.1 PROBLEM FORMULATION

Let's consider a Markov Decision Process [1] described by $\langle \mathcal{S}, \mathcal{A}, P, r^1, r^2, \gamma \rangle$ where:

- $\mathcal{S} = \mathcal{S}_1 \times \mathcal{S}_2$ is the set of joint environmental states constructed by the cartesian product of individual state sets $\mathcal{S}_i$ with cardinality $|\mathcal{S}| = |\mathcal{S}_1| \times |\mathcal{S}_2|$

- $\mathcal{A} = \mathcal{A}_1 \times \mathcal{A}_2$ is the set of joint actions

- $P = P(s'|s, a) : \mathcal{S} \times \mathcal{A} \to \Delta(\mathcal{S})$ describes the state transition dynamics of the environment, where $\Delta$ is a probability measure over $\mathcal{S}$.

- $r^i : \mathcal{S} \times \mathcal{A} \times \mathcal{S} \to \mathbb{R}$ is the reward function that returns a scalar value to the $i$-th agent for a transition from $s \in \mathcal{S}$, taking joint action $a \in \mathcal{A}$ to, $s' \in \mathcal{S}$.

- $\gamma \in [0, 1]$ is the discount factor for future steps.

The inverse reinforcement learning (IRL) problem (Abbeel & Ng, 2004)(Ziebart et al., 2008)(Ashwood et al., 2022) states that: given $\{P, \mathcal{S}, \mathcal{A}, \gamma\}$ and $N$ samples of expert trajectories $D = \{\zeta_1, \zeta_2, ..\zeta_N\}$, we aim to infer the unknown reward function $r^1, r^2$ in such a way that $P(D|r^1, r^2)$ is maximized. Each trajectory is composed of independent state-action pairs: $\zeta_i = \{(s_t, a_t)\}_{t=0}^T$. As a consequence, the posterior probability of observing expert trajectories $\zeta_i$ can be calculated with the specific policy $\pi$ derived from value functions (Eq. 1).

$$P(\zeta_i|r) = \prod_{t=0}^{T} \pi(a_t|s_t)p(s_t) \qquad (1)$$

Therefore, at the core of this problem is to parameterize the *joint policy function* $\pi(a|s)$ using a probabilistic modeling of social interactions based on MARL with recursive reasoning and value decomposition as highlighted in Figure 1.

### 3.2 VALUE FUNCTION PARAMETERIZATION AND DECOMPOSITION

To find the reward function $r : \mathcal{S} \to \mathbb{R}$ in a deterministic transition scenario, we need to infer $|\mathcal{S}|$ number of parameters. Such a formulation is computationally expensive in a multi-agent scenario, as the cardinality of $\mathcal{S}$ grows exponentially with the number of agents. To alleviate the issue, we adopted value decomposition such that a joint map could be decomposed into marginal maps and

---

[1]For simplicity, we provide equations for two agents. These derivations naturally extend to many agents albeit with exponential growth of required computation.

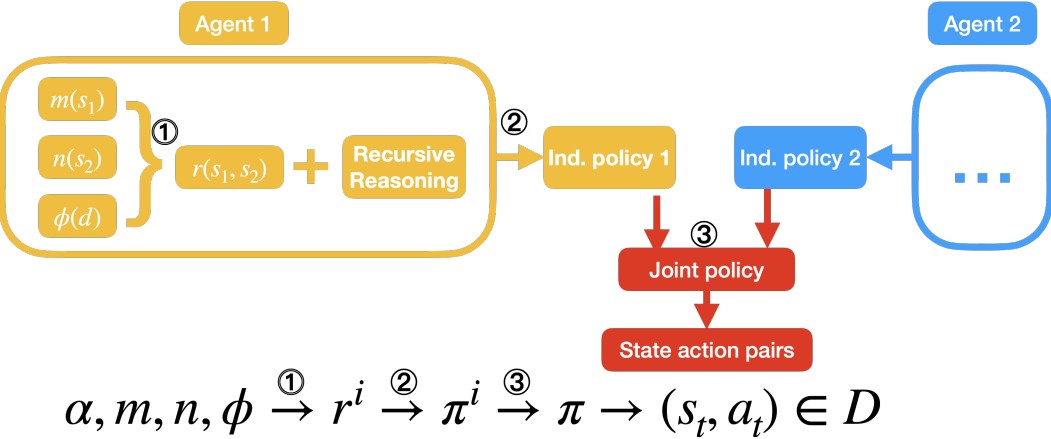

$$\alpha, m, n, \phi \xrightarrow{①} r^i \xrightarrow{②} \pi^i \xrightarrow{③} \pi \rightarrow (s_t, a_t) \in D$$

Figure 1: The forward generation graph of making social decisions. We observe state-space pairs and attempt to infer $m, n, \phi$ per agent. ①: value decomposition (Eq. 2), ②: policy generation with soft Q-learning (Eq. 3 and Eq. 4) and recursive reasoning (Table 1), ③: centralized or decentralized policy execution.

interactions maps as Eq. 2 where $s_i \in \mathcal{S}_i$ and $d$ stands for an given interaction function between $s_1$ and $s_2$ (e.g. Euclidean distance between $s_1$ and $s_2$ in a grid environment).

$$r(s_1, s_2) = \alpha_1 m(s_1) + \alpha_2 n(s_2) + \alpha_3 \phi(d(s_1, s_2)) \tag{2}$$

While the separability of the reward function $r$ is not always guaranteed, in Supplementary Section A.1 we demonstrate that, for multi-agent tasks with discrete state-action spaces, reconstruction error $r - m - n - \phi$ can be analytically calculated by solving an overdetermined linear system (Fig. S1). In the task of two-agent cooperative open arena foraging, we numerically evaluated the reconstruction error, which was found to be three orders of magnitude smaller than the reward strength (Fig. S2). When reward sparsity becomes larger, incorporating the interaction term becomes essential to maintain a low reconstruction error (Fig. S3). More interestingly, when three agents are considered in the task, we could still reconstruct the joint value function using their individual maps and two pairwise interaction terms with reconstruction error four orders of magnitude smaller (Fig. S4). These mathematical analysis results justified the validity of value decomposition and indicate the potential of our method to be applied to tasks with more than two agents.

At this stage, our objective is to learn the map weights $\alpha_k$, individual maps $m(s_1)$, $n(s_2)$ and interactions maps $\phi$ that maximize the posterior of the observed expert trajectories $D = \{(s_t, a_t)\}$.

## 3.3 POLICY PARAMETERIZATION

To facilitate the inference problem, we adopted a differentiable maximum entropy policy formulation (Ziebart et al., 2008)(Ziebart et al., 2010):

$$\pi(a|s) = \frac{\exp Q(s, a)}{\sum_{a' \in \mathcal{A}} \exp Q(s, a')} \tag{3}$$

where $Q(s, a)$ is a soft Q-function arising from performing soft value iteration:

$$Q(s, a) = r(s, a) + \gamma \sum_{s'} P(s'|s, a) \log\left(\sum_{a' \in \mathcal{A}} \exp Q(s, a')\right) \tag{4}$$

Note that a temperature term is not needed in the softmax function, as it will be absorbed into absolute value of $Q$, and subsequently $r$, which is the variable we aim to estimate.

In the multi-agent scenario, the above generative model only holds for a cooperative task where two agents have the same reward function and act in a centralized way. For non-collaborative tasks, we need separate $r^i$, $Q^i$ and $\pi^i(a|s)$ for each agent. Other cases involving decentralized policy execution will be discussed in the next sections.

## 3.4 LEVELS OF COGNITIVE HIERARCHY

The major complexity of modeling multi-agent interactions is the level of coordination and mutual predictions between agents. For notation simplicity, let's consider collaborative tasks where each agents have an identical value function. We further assume that agents know the state information of each other. A summary of the cognitive levels considered in this work was listed in Table 1.

In one extreme, agents generate an optimal joint policy $\pi(a_1, a_2|s)$ according to Eq. 3 and operate in a centralized way such that the optimal **joint action** would be executed. In this case, standard IRL formulations in Eq. 3 and 4 could be applied to calculate the posterior probability in an exponentially enlarged state space and action space. Note that in the following context, $\pi$ refers specifically to the optimal joint policy, not necessarily the policy adopted by agents.

In a less coordinated scenario, agents still generate an optimal joint policy $\pi$ but the policy execution process is independent, which is often the case in simultaneous action games. Mathematically, we have conditional independence of individual policies:

$$P(a|s) = P(a_1|s)P(a_2|s) \tag{5}$$

Then the question becomes how does each agent chooses its own action based on full state information. The answer lies in a probabilistic view of recursive reasoning.

For a level-0 or egocentric agent, who does not consider the action or state of other agents, its policy only depends on its own state information, i.e. $P(a_i|s) = P(a_i|s_i)$ and could be generated as soft value iteration of its own map function ($m$ or $n$).

For an agent with level-1 reasoning or higher, it builds a predictive model of the other player and is expected to act optimally based on this prediction. Let's denote agent 1's prediction of agent 2's policy as $\hat{a}_2$, based on which, we could break the policy of agent 1 as:

$$P(a_1|s) = \sum_i P(a_1, \hat{a}_2 = i|s) = \sum_i P(a_1|\hat{a}_2 = i, s)P(\hat{a}_2 = i|s) \tag{6}$$

Since a rationale agent 1 wants to act optimally for the joint task based on its own predictions, the first term is essentially the conditioned optimal policy:

$$P(a_1|\hat{a}_2, s) = \pi(a_1|a_2, s) \tag{7}$$

Thus the agent's prediction about its counterpart's policy as well as its own joint value map determines the formulation of its policy. The recursive level of this prediction also determines the agent's level of reasoning. For example, if agent 1 assumes agent 2 to be an egocentric, or level-0 agent (regardless of whether agent 2 is or not), then agent 1 has a cognitive level of one. If agent 2 assumes agent 1 is level-1 and forms its prediction about agent 1's policy (i.e. $P(\hat{a}_1|s)$) based on this assumption, agent 2 has a cognitive level of two. Even higher levels of reasoning can be constructed using this recursive process. Because most people do not reason beyond level 2 in strategic games, (Camerer et al., 2004)(Burnham et al., 2009)(Nagel, 1995), we begin by focusing on basic social interactions with low levels of social inference. Here we limit our consideration to two simple and extreme parameterizations of mutual prediction that provide the lower and upper bounds in terms of optimality in the case of independent control with prediction:

- If agent 1 has no information about agent 2's policy and generates chance level prediction (i.e. $P(\hat{a}_2|s) = |\mathcal{A}_2|^{-1}$), then $P(a_1|s) = |\mathcal{A}_2|^{-1} \sum_{a_2} \pi(a_1|a_2, s)$.

- If agent 1 predicts that agent 2 would act on the optimal marginal policy (i.e. $P(\hat{a}_2|s) = \pi(a_2|s)$), then $P(a_1|s) = \sum_i P(a_1|a_2 = i, s)\pi(a_2 = i|s) = \pi(a_1|s)$. In other words, agent 1 also acts on the marginal probability of the optimal joint policy.

## 3.5 INFERENCE PROCEDURE

The inference procedure is adapted from Ashwood et al. (2022) except that we have time-invariant version of weight estimation and different evaluation of joint policy given a chosen generative model for cognitive levels. Our objective is to learn the map weights $\boldsymbol{\alpha}$, individual maps $m(s_1)$, $n(s_1)$ and interactions maps $\phi(s_1, s_2)$ that maximize the posterior of observing expert trajectories $D = \{(s_t, a_t)\}$ under different generative models listed in Table 1.

Table 1: Generative models with different coordination levels.

| Generative model | Prediction model | Joint action probability |
|---|---|---|
| Centralized | / | $\pi(a_1, a_2\|s)$ |
| Independent control | egocentric agents | $P(a_1\|s_1)P(a_2\|s_2)$ |
| Independent control | chance prediction | $\prod_i \|\mathcal{A}_{-i}\|^{-1} \sum_{a_{-i}} \pi(a_i\|a_{-i}, s)$ |
| Independent control | optimal prediction | $\pi(a_1\|s)\pi(a_2\|s)$ |

---

**Algorithm 1** MAIRL with value decomposition

---

**Input1:** MDP information: $\langle \mathcal{S}, \mathcal{A}, P, \gamma \rangle$;
**Input2:** Expert trajectories $D$;
**Input3:** Hyperparameters: $K, \sigma_0, \sigma_1, \lambda_1, \lambda_2$; the chosen generative model in Table 1.;
Initialize $\boldsymbol{\alpha}, m, n, \phi$;
**for** $i = 1...N_{iter}$ **do**
    Calculate rewards $r$ as a function of $\boldsymbol{\alpha}, m, n, \phi$ (Eq. 2);
    Get *optimal joint policy* $\pi$ using softmax iteration until convergence (Eq. 3 and 4);
    Get *policy* $P(a\|s)$ based on the chosen generative model of cognitive level (Table 1);
    Update $\boldsymbol{\alpha}$ given reward map fixed (Eq.8, line 1);
    Calculate rewards $r$ and generate new policy as before;
    Update $m, n, \phi$ given weights fixed (Eq.8, line 2);
**end for**
**Output:** $\boldsymbol{\alpha}, m, n, \phi$

---

We assume all parameters have a Gaussian prior with known variance and mean and individual maps have different prior variance compared to the interaction map. Mathematically, $\boldsymbol{\alpha} \sim \mathcal{N}(\mathbf{1}, \Sigma)$ where $\text{diag}(\Sigma) = (\sigma_0^2, \sigma_0^2, \sigma_1^2)$, $m, n \sim \mathcal{N}(\mathbf{0}, \Sigma_m)$, $\phi \sim \mathcal{N}(\mathbf{0}, \Sigma_\phi)$. Note that we assume $\Sigma_m$ and $\Sigma_\phi$ are diagonal matrices. Incorporating the map prior is equivalent to adding an L-2 regularizer with coefficients $\lambda_1$ and $\lambda_2$ to the map entries. Mathematically, the parameter optimization progress could be written as:

$$\boldsymbol{\alpha}^* = \text{argmax}_{\alpha} \sum_{(s_t, a_t) \in D} \log P(s_t, a_t) - \frac{1}{2}((\boldsymbol{\alpha} - \mathbf{1})^T \Sigma^{-1}(\boldsymbol{\alpha} - \mathbf{1}))$$

$$(m^*, n^*, \phi^*) = \text{argmax}_{m,n,\phi} \sum_{(s_t, a_t) \in D} \log P(s_t, a_t) - \lambda_1 \|m\|^2 - \lambda_1 \|n\|^2 - \lambda_2 \|\phi\|^2$$

(8)

We performed coordinate ascent to iteratively update weights and maps while holding the other set of parameters fixed as illustrated in Alg. 1.

## 4 RESULTS

### 4.1 VALIDATION USING A SIMULATED COOPERATIVE FORAGING TASK

We first validated our method using simulations of a cooperative foraging task. In this task, two artificial agents navigate freely in an open arena of a five-by-five grid. On every trial, the two artificial agents are presented with two possible reward locations, but are only rewarded if they arrive simultaneously at the same target location (yellow squares in Fig. 2A). Both agents share the same ground truth reward function, which is defined over the joint state space and has non-zero values only at the states where both agents are at the reward locations. Simulation and fitting details can be found in the Supplementary.

#### 4.1.1 INFER MARGINALIZED VALUE MAPS AND TASK DEPENDENT INTERACTION MAP

The paths of two skilled agents (Fig. 2B and C) were simulated and analyzed using MAIRL with centralized learning and control. The resulting value maps perceived by both agents are shown in Fig.2D, E, and F. The target locations have higher value in the individual marginal goal maps, representing the available reward at these locations. Further, the map of Euclidean distance between the two agents (Fig. 2F) peaks when the agents are close to each other. This faithfully captures the

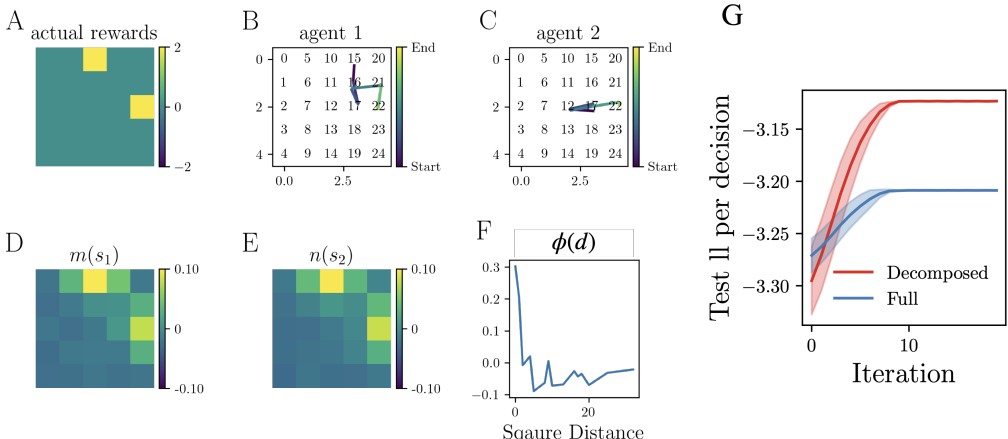

Figure 2: **Inference of decomposed value maps in a cooperative foraging task.** (A) In a five-by-five open arena, two artificial agents are only rewarded if they simultaneously arrive at the target locations (yellow cells). (B) (C) Example foraging trajectories of two expert agents starting from random positions and arriving at the east target location simultaneously. (D) (E) Estimated value map and weights ($\alpha$) of agent 1 or agent 2's position in the arena. (F) Estimated value function with respect to the agents' Euclidean distance in the arena ($\phi$). (G) Log likelihood (ll) of the test dataset along the iteration process of MAIRL with or without value decomposition. Each iteration step involves 40 full batch gradient descent steps.

key social element for solving this task: staying close to the partner. The larger $\phi$ compared to $m$ and $n$ also indicates that the agents place a higher subjective value on social proximity than being at the reward location. Importantly, the same inference algorithm without value decomposition resulted in a significantly lower data likelihood, likely getting stuck in a local minimum due to the excessive number of parameters to estimate (Fig. 2G). Thus, our MAIRL algorithm with value decomposition reveals the precise value each agents assigns to the two rewarding features in this tasks and invites further investigation of the neural representation of these values.

### 4.1.2 IDENTIFY LEVELS OF COGNITIVE HIERARCHY BY MODEL COMPARISON

Next, we infer the levels of cognitive hierarchy for the two agents using the simulated trajectories. The task lengths, represented by the number of foraging steps required to obtain rewards, appeared indistinguishable across models (Fig. S2). We compared models using Akaike Information Criterion (AIC) of the test trajectories. Conventionally, a drop of 2 or more in AIC is considered statistically significant. As shown in Fig. 3A, MAIRL effectively identified the ground truth model, except for close scores of the two independent control models when fitting the simulated chance prediction data (see limitations in Discussion). These findings underscore the efficacy of our approach in accurately inferring cognitive hierarchy from similar experimental trajectories.

In another simulation, where an egocentric agent was paired with a thoughtful level-1 agent, the task was still accomplished with a trajectory indistinguishable from that of a centralized pair (Fig. S2). By fitting MAIRL to the individual trajectories, we were able to identify the egocentric agent (Fig. 3B), but not the level-1 agent (Fig. 3C).

### 4.2 APPLICATIONS TO EXPERIMENTAL DATASETS

After validating in simulations where ground truth is available, we extended our approach to analyze two real datasets. The first dataset is collected from human participants performing a cooperative hallway task, and the second dataset is from monkeys performing a non-cooperative "chicken" game. By applying MAIRL, we recovered goal related value maps, task rule related interaction maps and the level of cognitive hierarchy, offering novel insights into the participants' subjective strategies and their social dynamics.

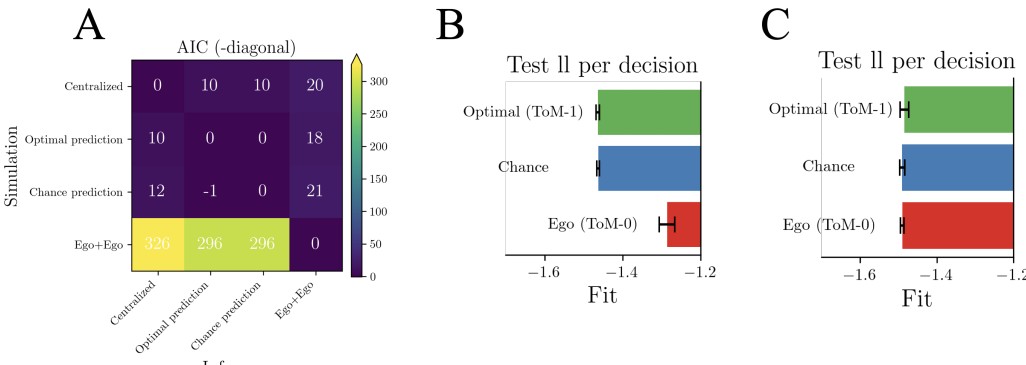

Figure 3: **Inference of cognitive hierarchy from similar trajectories.** (A) Akaike Information Criterion (AIC) of the test set simulated using the models listed on the vertical axis and fitted with the models listed on the horizontal axis. A decrease of 2 or more in AIC indicates a significantly better fit for the model. (B)(C) Log likelihood (LL) (mean ± std) of the test trajectories simulated for a pair of an egocentric agent and a level-1 agent. Trajectories of the egocentric individual (B) or level-1 agent (C) were fitted using generative models listed on the vertical axis.

### 4.2.1 INFER HUMAN PARTICIPANTS' STRATEGY AND TASK RULES IN A COOPERATIVE TASK

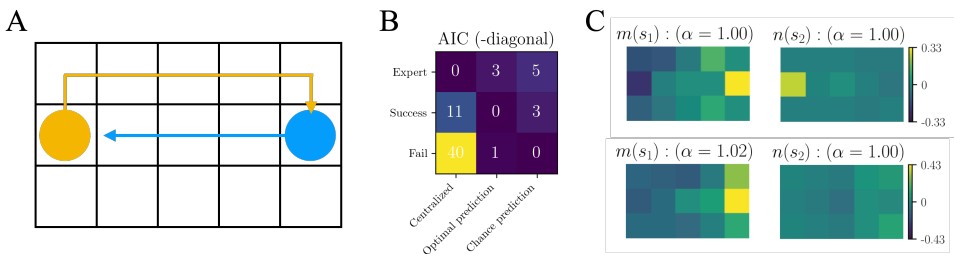

Figure 4: **Human participants' develop good estimation of each other to succeed for collaboration.** (A) Setup of the hallway task (B) Model fitness by AIC for trajectories of the three categories: fail, success and expert. (C) Estimated value maps of agent 1's trajectories from expert pairs (top) and failed pairs (bottom) using an optimal prediction model. In fail trials, the participant did not attach value to its counterpart's location ($s_2$). The value maps are similar when fitted using a chance prediction model (Fig. S3).

The hallway task(Ho et al., 2016) requires participants to start from orange or blue circles and switch locations without colliding. The original authors collected a total of 397 trials of human trajectories and categorized them into three groups: fail, success, and expert (completed with minimal number of steps), indicating increasing skill levels. Example trajectories are shown in Fig. S4.

We applied models of different cognitive levels to fit these trajectories (Fig. 4B). Pairs from expert trials exhibited remarkable coordination, akin to centralized control, despite the fact that each player was acting independently. This was evident from their capability for mutual predictions, as the value map of one agent is accurately predicted by the other (Fig. 4C). Pairs in successful trials displayed less coordination but maintained good mutual predictive capacity, supported by a well-fitted model of optimal prediction. For pairs in failed trials, trajectories were similarly explained by both optimal and chance prediction models. Nonetheless, inferred value maps using both models indicated that both participants did not consider each other's location (Fig. 4C and Fig. S3).

Unlike in cooperative foraging, where the key is to stay close to each other, the hallway task does not clearly indicate using Euclidean distance as an interaction term. Therefore, we tested different interaction maps, including Euclidean distance, column distance, row distance, and a joint map of both. In both expert and failed pairs, the model that used column distance as the interaction term

performed best (Fig.5A). The importance of column distance highlights the essence of the hallway task: switching column indices without colliding. Thus, states where the column distance equals zero (Fig.5C) are critical to solving the task. The recovered values showed that expert pairs highly valued these key states (Fig.5B). In contrast, failed pairs often collided and were bounced back to states where the column distance equaled one (Fig.5D), leading them to assign higher value to these states related to colliding (Fig. 5B).

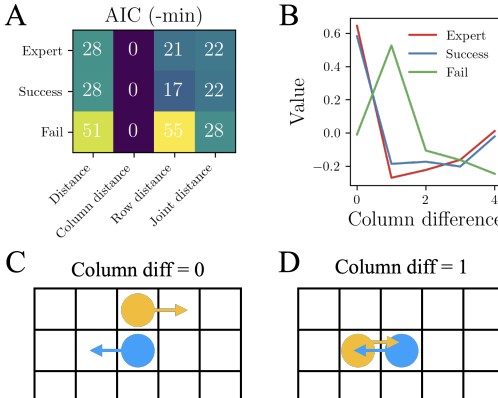

Figure 5: **MAIRL identified the most crucial interaction term related to the task setup**. (A) Model fitness measured by AIC for models with different interaction terms. The interaction map based on column distance performed best.(B) Value attached to the column distance between two agents. Expert trajectories prioritized states where the two agents pass each other (C), while failed trajectories focused on a column distance of one grid apart, suggesting potential collisions and rebounds (D).

### 4.2.2 INFER MONKEYS' MUTUAL PREDICTIONS IN A NON-COOPERATIVE TASK

The "chicken" game, a non-cooperative task blending coordination and conflict, offers insights into players' motivations and tendencies (Ong et al., 2021). Two monkeys are presented with options: go Straight (S) or Yield (Y), resulting in four potential outcomes (Fig. 6A and B):(i) S-S leads to a crash with zero reward; (ii)/(iii) S-Y yields a significant reward for the bold monkey and a small one for the yielding monkey; (iv) Y-Y results in cooperation with a moderate reward for both. Fig. 6B illustrates the outcome and payoff matrix from one monkey pair (trial number = 29,147), with decisions exhibiting roughly independent behavior, indicated by a mutual information measure of 0.01 bits.

This non-cooperative task is stateless with four actions in the joint action space. Two separate value functions, $V_1$ and $V_2$, need to be parameterized. The probability of agent 1's decision could be written as:

$$P_1(a_1) = \sum_{i \in \{S,Y\}} P_1(a_1|\hat{a}_2 = i)P_1(\hat{a}_2 = i) = \sum_{i \in \{S,Y\}} \pi_1(a_1|a_2 = i)P_1(\hat{a}_2 = i) \tag{9}$$

where $\pi_1(a_1, a_2)$ is a softmax transformation of $V_1(a_1, a_2)$ and denote $P_1(\hat{a}_2 = S) = q_1$. Inference details can be found in theSupplementary. The estimated $V_i$ and $q_i$ values (Fig. 6C) shows that Monkey 1 estimates that its opponent has a lower probability of going straight. Consequently, Monkey 1 is bolder and chooses to go straight more often. Interestingly, the behavior of this monkey matches its higher social status as determined by a separate confrontation experiment.

Notably, the authors of these experiments (Ong et al., 2021) developed a "hybrid RL and strategic learning model" to determine one animal's prediction about another, which is equivalent to a constrained MARL model with independent control, mutual logistic prediction and hard Q-learning update. Compared to the this model, our MAIRL model are more generally applicable to more complex predictions in social interactions and state-dependent behaviors.

## 5 DISCUSSION

We used MARL and recursive reasoning to model goal-directed reward representation and levels of cognitive hierarchy levels in social tasks and applied IRL to infer value functions from expert trajectories and determine the level of reasoning by model comparison. We validated our new approach using both simulated tasks and experimental datasets. Unlike previous methods, our

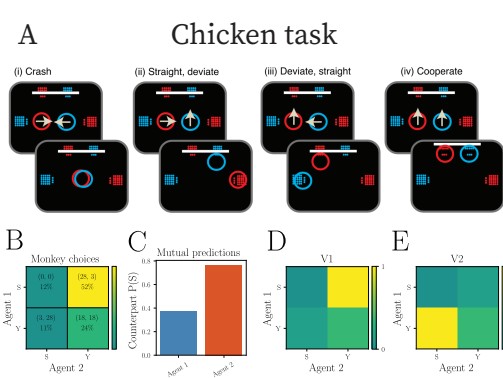

Figure 6: **Monkey's mutual predictions are related to their social hierarchy in a non-cooperative task.** (A) Two monkeys (red and blue circles) choose to go Straight (S) or Yield (Y) in a non-cooperative chicken task. (adapted from (Ong et al., 2021)) (B) The choices of a pair of monkeys from the experiment (Fig. 2A in (Ong et al., 2021)). The value pairs in parenthesis indicate the pay-off matrix. (C) (D) (E) Estimated mutual predictions and value maps using a generative model of independent control with prediction. Mutual predictions are the probability of Monkey 2 choosing S as predicted by Monkey 1 and vice versa. Monkey 1 chooses to go straight more often because it is unlikely for Monkey 2 to go straight. This strategy matches their social hierarchy where Monkey 1 has a higher rank than Monkey 2.

approach disentangles agents' perception of task-related goal values from their recursive reasoning about others and provides an interpretable estimation of agents' value function.

This work presents a compelling adaptation of MARL theories to model social interactions, propelling research in neuroscience and cognitive sciences. By integrating advanced machine learning techniques and theoretical guarantees, we provide novel insights into the computations underlying complex social dynamics.

**Limitations.** We currently assume that each agent has full access to the entire state information. In the foraging task, this implies that each agent knows the locations of all other agents, which may be an unrealistic assumption. A more plausible approach would involve adopting a multi-agent POMDP framework, where agents' locations are hidden and can only be inferred through field-of-view observations. The current algorithm generates joint policies using the joint value function, resulting in computational complexity that grows exponentially with the number of agents. Decentralized policy evaluation could potentially be implemented in the future by assuming a Nash equilibrium (Reddy et al., 2012) or other solutions to improve computational efficiency for scenarios involving more than two agents. Nonetheless, whether such decentralized methods can still preserve the differentiable computation graph remains uncertain and requires further investigation. Our algorithm also assumes a non-parametric reward function in discrete state and action spaces. A potential improvement would be to adopt a parametric reward function (e.g. a diffusion function like in (Ullman et al., 2009)) in continuous state and action spaces, which may enhance efficiency. Beyond limitations in modeling assumptions, our current comparison did not reveal significant differences between the two predictive models under independent control (Fig. 3A), potentially because the task allows agents to perform effectively by making basic predictions about the other's actions. To explore the framework's full potential, it could be extended to more complex social scenarios that demand advanced mentalization processes, such as the OverCooked task. (Wu et al., 2021).

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

## A    SUPPLEMENTARY DETAILS

### A.1    ANALYSIS OF VALUE DECOMPOSITION

We assumed that joint value function $r(s_1, s_2) : \mathcal{S} \to \mathbb{R}$ can be approximated by decomposing it into individual value maps $m(s_1) : \mathcal{S}_1 \to \mathbb{R}$, $n(s_2) : \mathcal{S}_2 \to \mathbb{R}$ and an interaction term $\phi(d(s_1, s_2))$ as a function of Euclidean distance $d(s_1, s_2)$. While this separability property is not universally achievable, we demonstrate that, for multi-agent tasks with discrete state spaces, reconstruction error $r - m - n - \phi$ could be analytically calculated by solving an overdetermined linear system.

Formally, we want to approximate a fixed value function $V(s_1, s_2)$ with a summation term $m(s_1) + n(s_2) + \phi(d(s_1, s_2))$. This poses $m \sim O(|\mathcal{S}|^2)$ constrains to a system with $p \sim O(|\mathcal{S}|)$ parameters, constructing an overdetermined linear system. As depicted in Fig. S1, we could write down the reconstruction equation (Eq. 2) as $y = X\beta$ where $y$ is a flattened version of $V(s_1, s_2)$, $\beta$ is a concatenated vector of parameters in the $m, n, \phi$ and $X$ is a fixed design matrix depending on the functional form of $d$ and the grid environment. Since $m > p$, this is an overdetermined system, we could write down the ordinary least square (OLS) solution as $\hat{\beta} = (X^T X)^{-1} X^T y$ and the error vector evaluated as $(I - X(X^T X)^{-1} X^T)y$, denoted as $My$. The reconstruction error is the projection of vector $y$ onto the null space of $M$. Since $M$ and $y$ both depends on the task setup and the choice of the interaction term, we could pre-calculate the reconstruction error before performing inference.

For the two agent cooperative foraging task, the reconstruction error was inspected in Fig. S2. In this example, we assigned a reward strength of 1 when both agents are at location $(0, 0)$ and the reconstruction error is at the order of $10^{-3}$ for a grid size of 5, which is negligible. The OLS solution of $m$ and $\phi$ were plotted in the middle and right panel of Fig. S2. The individual map highlighted the reward location at $(0, 0)$ and the interaction map emphasizes the importance of being together (i.e. $\phi(0)$). Note that reconstruction error decreases as the number of grid size increase because $y$ gets sparser in larger arenas. Actually, when reward is not sparse, the interaction term is essentially in achieving minimal reconstruction error (Fig. S3). The two agent hallway task has similar formulation except a different reward location in vector $y$. Therefore, we could also achieve minimal error using value decomposition.

More interestingly, we could also evaluate the decomposition performance for a three-agent cooperative foraging task. In this task, agents will only be rewarded if all of them are at the same reward port simultaneously. The reward ports are placed on the diagonal of a $5 \times 5$ square arena. Value decomposition is computed as $V(s_1, s_2, s_3) = m(s_1) + n(s_2) + l(s_3) + \phi_1(d(s_1, s_2)) + \phi_2(d(s_1, s_3))$. Reconstruction error is at the order of $10^{-4}$ and the OLS solution also indicates the position of reward ports and the value of staying together for an agent pair. This result indicates the potential of applying our method to tasks with more than two agents.

### A.2    SIMULATION DETAILS OF THE COOPERATIVE FORAGING TASK

The arena contains a 5-by-5 grid with deterministic action transitions, resulting in 625 states and 25 actions in the defined two-agent MDP. Two terminal states are defined by the simultaneous presence of both agents at the same reward location. If an agent hits the boundary, only its own state is reset to the previous time step. The reward strength of two locations was sampled from $\mathcal{N}(2, \sigma_0^2)$ with $\sigma_0 = 0.01$. The average reward strength was handpicked to be 2 to enable a balance between sampling from all state-action pairs and emphasizing the importance of centralized control.

At the beginning of each trial, both agents are placed randomly in the arena. Their exploration policy was generated using value iteration and the chosen policy execution model from Table 1. For all datasets, 500 trials were simulated with a maximum length of 200 steps. The discount value $\gamma$ was set to 0.9.

### A.3    INFERENCE DETAILS FOR COOPERATIVE FORAGING

We used 80% trajectories for model fitting and the other 20% to calculate the test log-likelihood as model evidence. The preset hyperparameters include: the future discount factor $\gamma = 0.9$ and the

covariance strength of map weights $\sigma_0 = \sigma_1 = 0.01$. In this way, we're imposing a strong prior of the map weights to be around one, for a better comparison of the recovered maps.

Other hyperparameters were searched among the range provided in Table S1 in a centralized dataset inferred using centralized model assumptions. We picked $\eta_\alpha = 0.01, \eta_{map} = 0.005, \lambda_1 = 5, \lambda_2 = 1$ based on training stability, model evidence and map interpretability.

| $\eta_\alpha$ | learning rate for weights | [0.001, 0.005, 0.01] |
|---|---|---|
| $\eta_{map}$ | learning rate for the map | [0.001, 0.005, 0.01] |
| $\lambda_1$ | L2-penalty (prior) for individual maps | [0,1,5] |
| $\lambda_2$ | L2-penalty (prior) for the interaction map | [0,1] |

Table S1: Range of hyperparameters tested

We initialized the weights to be one, the individual maps to be uniformly sampled from 0 to 1, and the location difference map to be $1/(d+1)$ where $d$ is the square Euclidean distance between two agents. We used Adam optimizer with learning rate given above for iterate for 400 epochs. Most loss curves would plateau around 200 epochs. Example loss curves for the training and test set were shown in Fig.S5.

For 500 expert trajectories with approximately 30 decisions per trajectory, the inference process took around 5 minutes on a single CPU thread (Intel(R) Xeon(R) CPU E5-2687W 0 @ 3.10GHz).

### A.4 INFERENCE DETAILS FOR THE HALLWAY TASK

For the hallway task, most hyperparameters are the same as the cooperative foraging task except that we searched for $\lambda_1 \in [0,1,2,5]$ and $\lambda_2 \in [0,1,2]$ using the success trajectories. We used $\lambda_1 = 2$ and $\lambda_2 = 1$.

### A.5 INFERENCE DETAILS FOR THE CHICKEN TASK

Unlike the previous two tasks, this non-cooperative task is stateless with four actions in the joint action space. Two separate value functions, $V_1$ and $V_2$, need to be parameterized. Value decomposition was not used due to limited number of parameters to estimate. The probability of agent 1's decision could be written as:

$$P_1(a_1) = \sum_{i \in \{S,Y\}} P_1(a_1|\hat{a}_2 = i)P_1(\hat{a}_2 = i) = \sum_{i \in \{S,Y\}} \pi_1(a_1|a_2 = i)P_1(\hat{a}_2 = i) \qquad (10)$$

where $\pi_1(a_1, a_2)$ is a softmax transformation of $V_1(a_1, a_2)$ and denote $P_1(\hat{a}_2 = S) = q_1$. Therefore, $P_i(a_i)$ is parameterized by five parameters each while the observed choice probability only offers two observation points. This creates an under-determined system. So we posed some constrains over the value function according to the task setup. We constrained $V_1$ to be the transpose of $V_2$ and $V_1(S,Y) > V_1(Y,Y) > V_1(Y,S) > V_1(S,S) = 0$ according to the task's nature. In this way, we could have a faithful estimation of $q_1$ and $q_2$, which is usually the biggest ambiguity in these social interactions.

## B SUPPLEMENTARY FIGURES

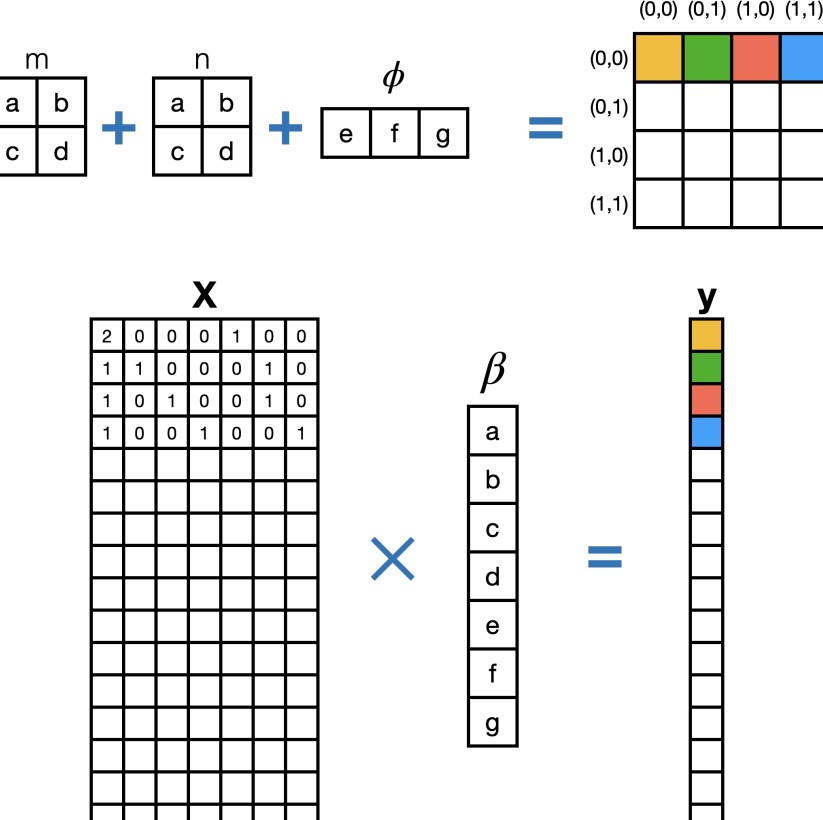

Figure S1: **Value decomposition as an overdetermined linear system.** For illustration purpose, this task includes two agents interacting in a $2 \times 2$ square arena with square grid distance as the interaction term $\phi$. Without loss of generality, we could assume $m$ and $n$ are the same. We have seven free parameters $a := m(0,0), b := m(0,1), c := m(1,0), d := m(1,1), e := \phi(0), f := \phi(1), g := \phi(2)$ to reconstruct 16 value functions on the right. We could write down the design matrix $X$ based on the relationship between states and the arrangement of the grid environment. Examples are given for the first 4 colored joint states.

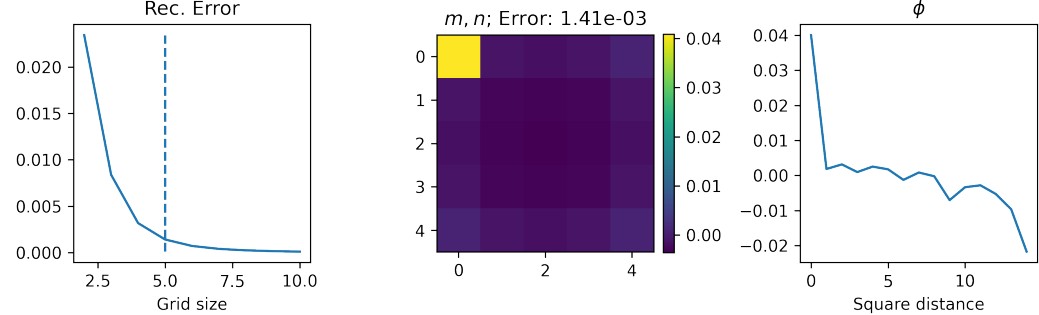

Figure S2: **Value decomposition for two agent cooperative foraging** Left: Mean square error of value reconstruction with respect to arena grid size, using one individual map (blue) or two individual maps (orange) plus the interaction term. Middle and Right: When the arena is $10 \times 10$ and the target location at $(0, 0)$, the OLS solution of individual maps and interaction maps.

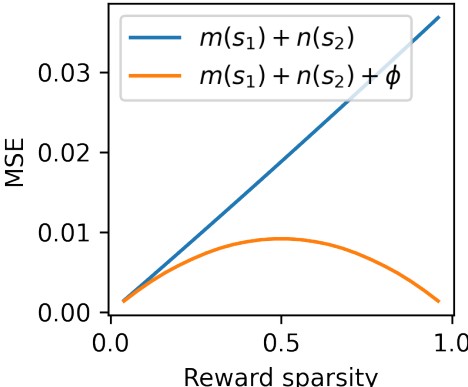

Figure S3: **Reconstruction error with or without interaction map.** When reward ports increases in two agent cooperative foraging in a $5 \times 5$ open arena, the interaction term is necessary to achieve good reconstruction.

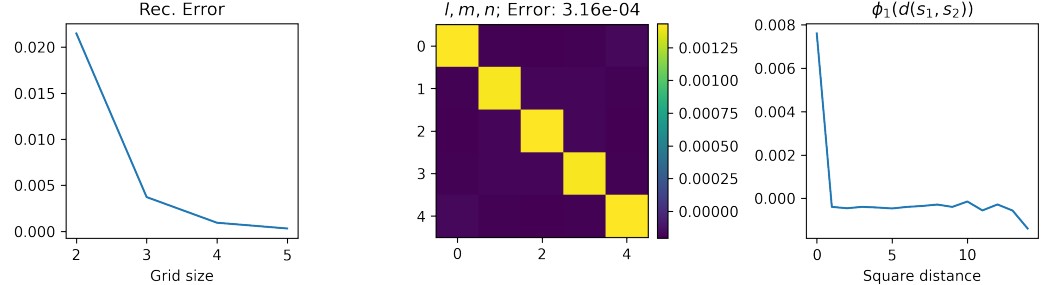

Figure S4: **Value decomposition for three agent cooperative foraging** Upper left: mean square error of valuer reconstruction using two interactions maps (blue) and three interactions maps (orange) with respect to arena grid size. Two interaction terms are enough to reconstruct the joint value function when the arena gets bigger; When the arena is $5 \times 5$, the OLS solution of individual maps and two interaction maps.

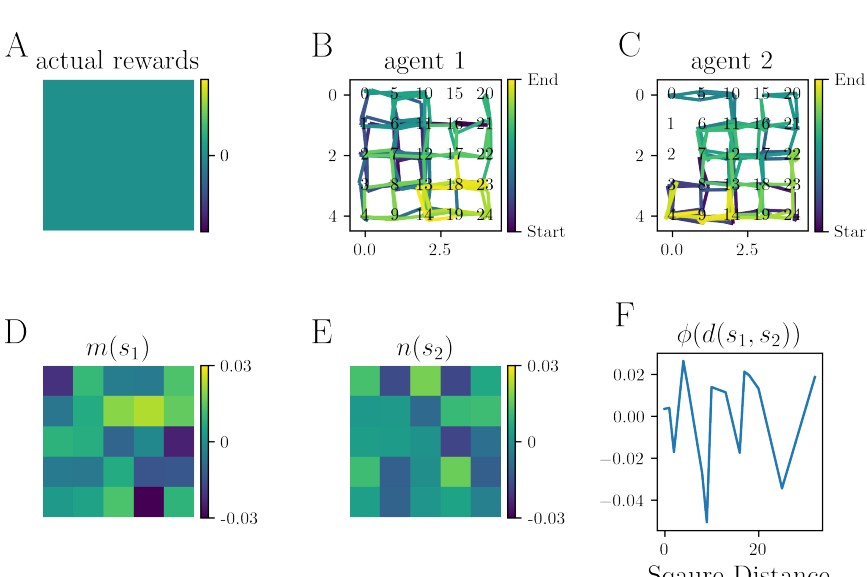

Figure S5: **Value recovery in random walk**

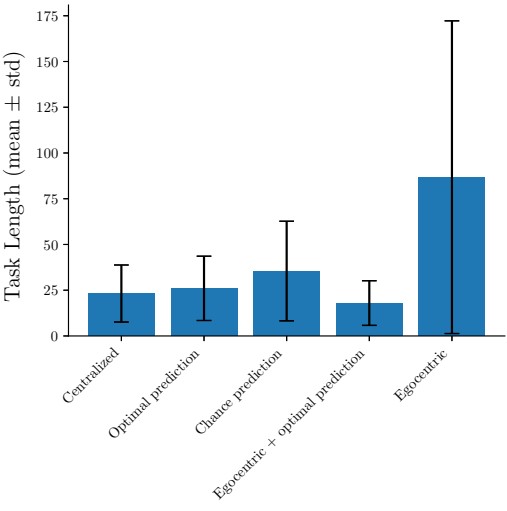

Figure S6: **Task length simulated using different models**

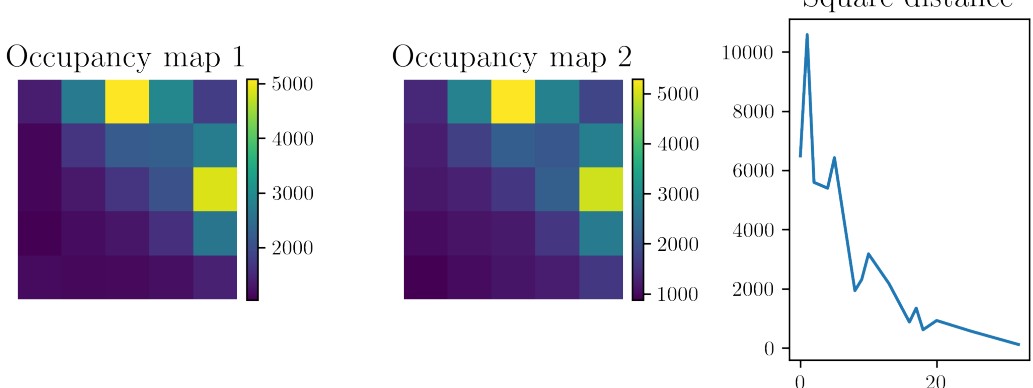

Figure S7: **Occupancy map for agents simulated using independent control with optimal prediction.**

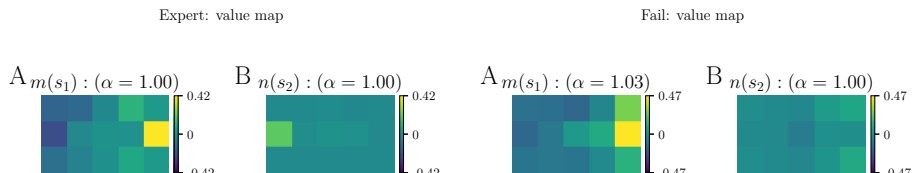

Figure S8: **Expert and failed trajectories fit using independent control with chance prediction**

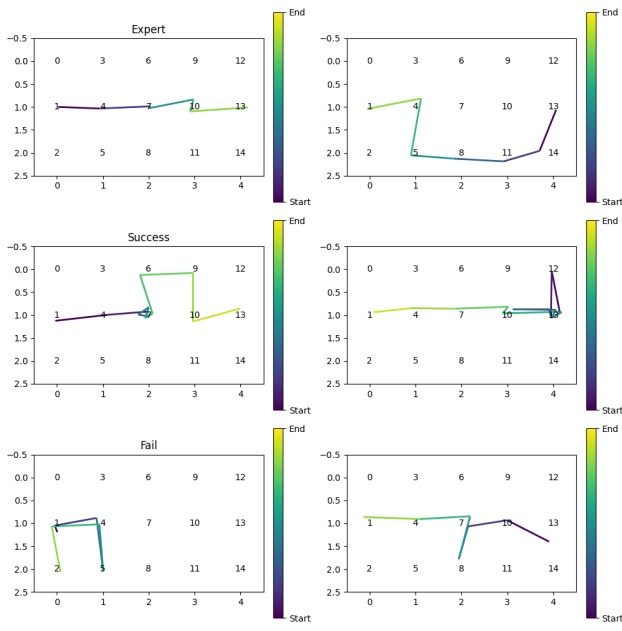

Figure S9: **Example expert (upper), success (middle) and fail (bottom) trajectories in the hallway task**

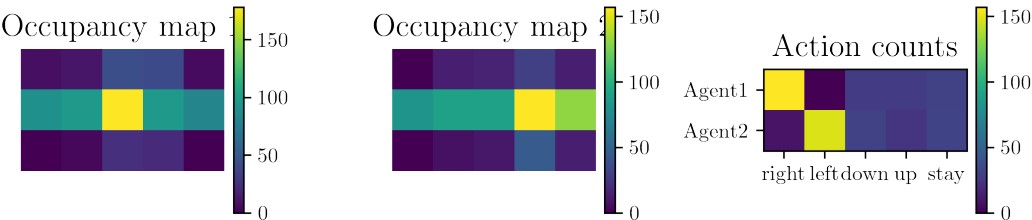

Figure S10: **Occupancy map for expert pairs in the hallway task**

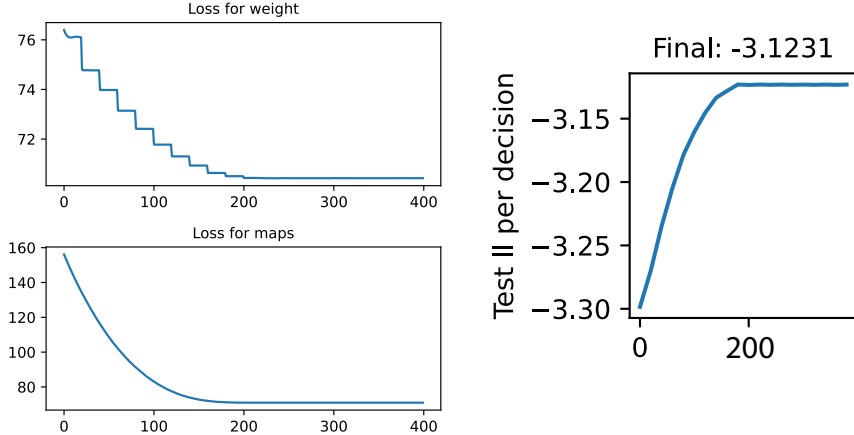

Figure S11: **Example loss curves for a centralized dataset inferred using centralized model assumptions.** Upper left: Loss (LL+prior) curve for weights. The curve is step-wise because we alternated optimizing for weights and maps and there is little room for weight change. Lower left: Loss (LL+L2 penalty) for maps. The curve plateaued around 200 epochs. Right: Test LL per decision along training.

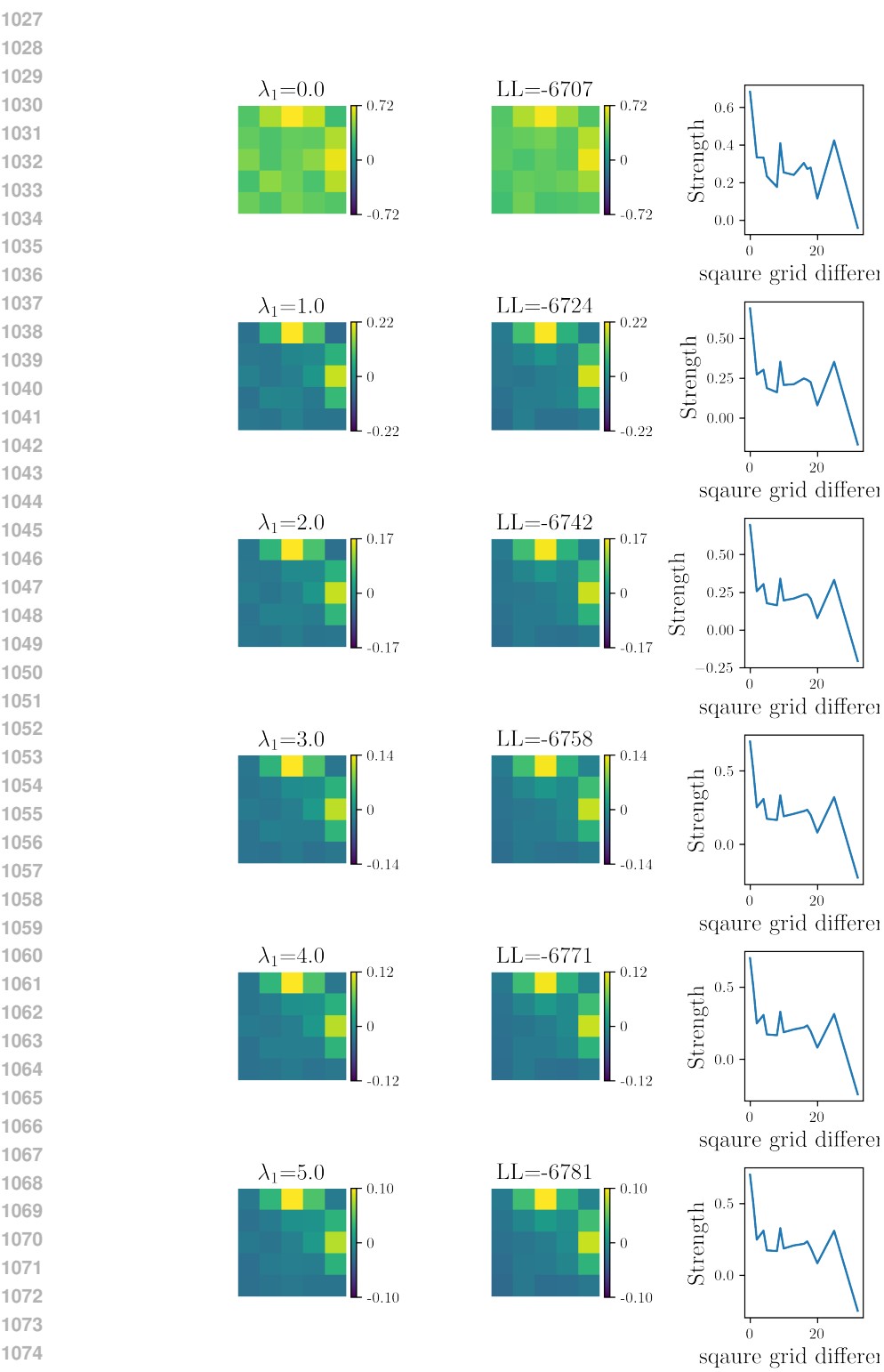

Figure S12: **Value map recovery with different choice of** $\lambda_1$

