# OpenReview forum: "Unveiling the latent dynamics in social cognition with multi-agent inverse reinforcement learning"
_ICLR.cc/2025/Conference — Submitted to ICLR 2025_

### Official Review · Reviewer_85mY · 2024-10-24

**Soundness:** 2
**Presentation:** 2
**Contribution:** 3
**Rating:** 6
**Confidence:** 4

**Summary:**

This paper is about multi-agent inverse reinforcement learning (MAIRL). It applies an existing method for IRL to a multi-agent context in order to infer the extent to which agents take each other's goals into account in social interactions. By decomposing the joint policy of two agents in different ways, different hypotheses about the extent to which each agent takes the other agent's behavior into account. The method is validated on a simulated foraging task in a grid world and then applied to two behavioral data sets: one cooperative task with human participants and one non-cooperative task with monkeys.

**Strengths:**

The application of MAIRL in the context of cognitive science as an inference method from a researcher's perspective to estimate the costs of two agents seems like a substantially novel contribution to me. To my knowledge, most of the other work in cognitive science on IRL in a multi-agent context seems to be on modeling theory of mind of the individual agents, but not on using MAIRL as a tool to analyze collaborative behavior. The central new methodological trick of the paper is the decomposition of the reward function into the rewards of the individual agents and a collaborative term, which makes inference in a multi-agent IRL setting more tractable. The conceptual framework is appealing and the applications to behavioral data promise to yield potentially interesting insights about the extent to which agents collaborate. I also appreciate that the code is available as supplementary material.

**Weaknesses:**

I have two main criticisms about this paper. First, the discussion of related work seems incomplete and at times confusing. Second, the evaluation of the method suffers from the fact that it was only evaluated on a single task, of which the authors say that it is a "simple task [that] does not require a sophisticated mentalization process", which raises doubts about the central claim that the method infers something about the "levels of cognitive hierarchy". I expand on these points in detail below.

1. The discussion of related work in the introduction and related work section seems quite superficial. It makes it seem like there is no prior work on inverse reinforcement learning in a multi-agent setting. While I see the novelty of the proposed application of MAIRL as an inference method from the perspective of a cognitive science researcher, there is relevant technical work on MAIRL and other applications of IRL in a multi-agent setting in cognitive science.

   a. There is no discussion of how the proposed approach compares to any other technical work on MAIRL or IRL in multi-agent settings. Some references that seem relevant from a cursory search:

    - Natarajan, S., Kunapuli, G., Judah, K., Tadepalli, P., Kersting, K., & Shavlik, J. (2010, December). Multi-agent inverse reinforcement learning. In *2010 ninth international conference on machine learning and applications* (pp. 395-400). IEEE.
    - Waugh, K., Ziebart, B. D., & Bagnell, J. A. (2011). Computational rationalization: the inverse equilibrium problem. In *Proceedings of the 28th International Conference on International Conference on Machine Learning* (pp. 1169-1176).
    - Reddy, T. S., Gopikrishna, V., Zaruba, G., & Huber, M. (2012, October). Inverse reinforcement learning for decentralized non-cooperative multiagent systems. In *2012 ieee international conference on systems, man, and cybernetics (smc)* (pp. 1930-1935). IEEE.
    - Rabinowitz, N., Perbet, F., Song, F., Zhang, C., Eslami, S. A., & Botvinick, M. (2018, July). Machine theory of mind. In *International conference on machine learning* (pp. 4218-4227). PMLR.

    b. Also, almost no reference is made to other work from cognitive science or behavioral economics about goal-inferences in multi-agent scenarios. Some references that seem relevant:

    - Wu, S. A., Wang, R. E., Evans, J. A., Tenenbaum, J. B., Parkes, D. C., & Kleiman‐Weiner, M. (2021). Too many cooks: Bayesian inference for coordinating multi‐agent collaboration. *Topics in Cognitive Science*, *13*(2), 414-432.
    - Carroll, M., Shah, R., Ho, M. K., Griffiths, T., Seshia, S., Abbeel, P., & Dragan, A. (2019). On the utility of learning about humans for human-ai coordination. *Advances in neural information processing systems*, *32*.
    - Kuleshov, V., & Schrijvers, O. (2015). Inverse game theory: Learning utilities in succinct games. In *Web and Internet Economics: 11th International Conference, WINE 2015, Amsterdam, The Netherlands, December 9-12, 2015, Proceedings 11* (pp. 413-427). Springer Berlin Heidelberg.

2. The point in the related work section, which discusses drawbacks of POMDPs and introduces MDPs as the favored model raised multiple questions for me.

    a. "While fruitful, this [POMDP] approach has several drawbacks. First, there is a lack of consistent mapping between the inferred value function and the specific mental state of the agent such as goal." Why is there a "lack of consistent mapping" in the POMDP approach, and why is this not the case in MDPs? Please explain.

    b. Confusingly, the papers cited as references for the POMDP approach (Baker et al., 2009; Velez-Ginorio et al., 2017) do not even use POMDPs, but seem to model inverse planning in an MDP setting.

    c. More generally, modeling a situation, which actually involves partial observability, with a model that does not properly take partial observability into account might lead to wrong inferences about their goals (see e.g. Straub, Schultheis, Koeppl & Rothkopf, NeurIPS 2023). If an actual task involves partially observability and therefore is described well by a POMDP, why would it be advantageous to use an MDP instead?

3. John Maynard Keynes' (1937) "general theory of employment" is cited as a reference for the claim that "even human reasoning has a depth of one or two levels". To me, the depth of recursive theory of mind in humans seems like an empirical question. While Keynes might be motivated by assumptions about human cognition, I am not aware that he performed behavioral studies investigating this question. Please clarify.

4. The only example on which the algorithm is evaluated using simulated data is the "cooperative foraging task" in Section 4.1. At the end of the section, the authors state that it was not possible to distinguish the "level-1 theory of mind agent" from a "chance prediction agent" given the data. The suspected reason for this is that "this simple task does not require a sophisticated mentalization process" (l. 347). This is in contrast to the claim in the introduction that the method estimates the "levels of cognitive hierarchy between two agents". In light of this evaluation, we cannot be sure that the inference method works for more complex tasks that might require "sophisticated mentalization processes".

Minor points:
- "Note that a temperature term in not needed [...]" should be "is" (l. 186)
- Both the distribution of reward strengths in the simulation (Supplementary A.1) and the prior over map weights are called $\sigma_0$.
- "uniformed sampled" should be "uniformly sampled" (l. 680)

**Questions:**

1. Why is the "egocentric agents" model not included in the model comparisons (Fig. 3A, Fig. 4B)?
2. The parameter $\beta$ in Eq. (2) does not show up again anywhere in the paper. Does it have the same function as the map weights that are later called $\alpha$?
3. On hyperparameter selection: "We picked $K = 1, η_α = 0.01, η_{map} = 0.005, λ_1 = 5, λ_2 = 1$ based on training stability, model evidence and map interpretability." (l. 670) - What does it mean to pick the hyperparameters based on map interpretability? How was the model evidence traded off against this subjective criterion?
4. The inference of the column distance as the interaction map for the hallway task in Section 4.2.1 raises two of questions.
    a.  While the column difference indeed distinguishes the states in Fig. 5C and 5D, the row difference is also different between the two plots. Would a model that maximizes row distance instead of minimizing column distance not explain the behavior?
    b.  While Fig. 4 lists "expert", "success", and "failed" trials, Fig. 5 only shows "expert" and "failed" trials. Why?

---

> ### Author Response · Authors · 2024-11-27
> **Response to Reviewer 85mY**
>
> We thank the reviewer for great suggestions. We uploaded a revised pdf with changes highlighted in blue and addressed the reviewer's concerns point by point:
>
> **Related work.**  We did another round of literature review and rewrote the related work session. Out of the references the reviewer mentioned, Rabinowitz et al 2018 seems irrelevant because the proposed ToM-net architecture seeks to maximize predictability of other agent’s actions and doesn’t explicitly infer a reward function. Carrol et al. incorporated behavioral cloning (i.e. a supervised classifier to predict action) to aid in human-computer interaction, which is irrelevant to inverse RL.
>
> **POMDP v.s. MDP** We apologize for the confusion about references. Baker et al 2017 used POMDP while Baker et al 2009 and Velez-Ginorio 2017 use MDP. We revised them in the new paragraph. The ‘While fruitful…’ sentence was paraphrased from the review paper by Jara-Ettinger in 2019 (Fig. 1 legend). It might be confusing to mention it here without context. We deleted this as well as the comparison of MDP and POMDP.  As noted in the revised ‘related work’ part, most cognitive sciences work adopted single agent POMDP and multi-agent MDP work to model social interactions. We chose to use multi-agent MDP in our work and look forward to adopting it into a multi-agent POMDP scenario.
>
> **John Maynard Keynes' (1937) "general theory of employment" is cited...** The beauty contest in game theory is a behavioral finance concept that describes how people make predictions based on what they think others will do, instead of the intrinsic value of an asset. The game is named after John Maynard Keynes, who used the metaphor in his book to explain price fluctuations in the stock market. Burnham et al. (2009) conducted a Keynes beauty contest in 658 subjects and reported value choice from 17 to 50, indicating that the majority of participants exhibited a cognitive reasoning level of approximately zero or one. We have updated the references accordingly in the manuscript.
>
> **We cannot be sure that the inference method works for more complex tasks...** We appreciate the reviewer’s insightful comments. First, we would like to clarify that our algorithm successfully distinguishes between level-1 and level-0 agents. Level-1 agents make predictions about the other agent’s behavior, while level-0 agents act non-strategically but still pursue self-serving actions using the basic features of the environment. This distinction demonstrates the algorithm's ability to infer differing reasoning capabilities. However, we acknowledge that the model is currently unable to distinguish between two types of level-1 agents that employ different strategies for predicting the other agent's behavior.  Regarding the statement, "this simple task does not require a sophisticated mentalization process," we meant that the cooperative foraging task may not demand nuanced or complex predictions about the other agent. Importantly, our results indeed revealed that mentalizing at level-1 of the cognitive hierarchy is critical for success in this task. To prevent any confusion, we have now removed the aforementioned sentence and revised the section accordingly.
>
> **Egocentric agents.** Egocentric agents perform so differently from other kinds of agents because it takes them significantly more steps to perform the cooperative task (Fig S6). Therefore, we could distinguish these agents even without modeling fitting. We still updated the fit results of egocentric models to Fig.3A. They are substantially worse.
>
> **Eq. (2)** It should be $\alpha$. We modified Eq.2
>
> **Hyperparameter selection.** We first tested the learning rate of 0.001, 0.005, 0.01, 0.05. Based on convergence rate and  training stability, we picked 0.01 for weight optimization and 0.005 for value map optimization. Fixing these two learning rates and $\lambda_2=1$, we tested $\lambda_1$ from the pool of 0, 1, 5, 10, 20, 30 and got the model evidence curve with respect to this hyperparameter. The log likelihood decreases when $\lambda_2$ increases and the recovered map looks similar for any $\lambda_2>0$ (Fig.S12). We picked $\lambda_1=5$ but also think this choice will not affect most conclusions in the paper. Note that we only care more about the ratio between $\lambda_1$ and $\lambda_2$ because the value map is robust to scaling.
>
> **Column distance and row distance** In the hallway task, two agents cannot occupy the same grid. Agent 1 (on the left) tends to move to the right, while Agent 2 tends to move to the left (Fig.S10). A state where the column distance is zero indicates that the two agents are on different rows and successfully passing each other. However, a row distance of one can also occur when the agents are still far apart column-wise, which does not necessarily ensure successful task completion.
>
> **Fig. 5** We added the fitting results for ‘success’ trials in Fig.5. Their results look very similar to ‘expert’ pairs.

---

> > ### Comment · Reviewer_85mY · 2024-12-02
> >
> > I appreciate the additional related work that is now cited and the clarification of POMDPs vs MDPs in response to my review.
> >
> > I am still unsure about the results in Section 4.1.2 "Identify levels of cognitive hierarchy by model comparison". As far as I can understand, the method does not seem to be able to distinguish between an agent that assumes that the other agent acts completely randomly (chance prediction) and an agent that assumes the other agent acts optimally (optimal prediction). The original version of the article stated that this is not due to the method but due to the task that was chosen as an evaluation ("this simple task does not require a sophisticated mentalization process"). In response to my review, the authors have chosen to simply remove the sentence in question, instead of further explaining it. Why was the method not evaluated on a task that would "demand nuanced or complex predictions about the other agent"?

---

### Official Review · Reviewer_KWEN · 2024-10-31

**Soundness:** 2
**Presentation:** 3
**Contribution:** 2
**Rating:** 5
**Confidence:** 3

**Summary:**

This paper presents an approach to modeling social behavior in multi-agent settings, leveraging ToM principles through a probabilistic recursive model of cognitive hierarchy and joint value decomposition. By applying multi-agent IRL (MAIRL), the authors aim to uncover agents' latent beliefs and intentions in complex social environments. They validate this model in both cooperative and non-cooperative settings, demonstrating its potential to reveal meaningful behavioral insights without explicit task rules.

The integration of cognitive hierarchy and value decomposition in multi-agent inverse reinforcement learning is a compelling approach to understanding social behavior.

The approach seems to be generally sound and well evaluated, and so, I mainly have a few fundamental questions. Please note, some of my questions are merely exploratory, but I'd appreciate your thoughts:
- It wasn't immediately clear to me that how do you determine the appropriate level of cognitive hierarchy to model for each agent, especially in scenarios with varied agent capabilities? For example, different agents may interpret or respond to social cues at different levels.

- Given the known computational load in multi-agent recursive reasoning (increased k in k-level reasoning), how does the computational cost of your approach scale with an increasing number of agents or cognitive levels?

- Where do the trajectories needed for inverse multi-agent reinforcement learning are assumed to come from? Reliable and representative trajectory data are essential, and more insights would be very helpful, particularly given the challenges of collecting expert data from humans in multi-agent settings (see [1] for instance).

[1] Seraj et al “Mixed-initiative multiagent apprenticeship learning for human training of robot teams”, NeurIPS 2023

- How can we be confident that the "latent beliefs" uncovered by your MAIRL model truly represent the agents' underlying beliefs, rather than merely approximating observed behaviors? Can you shed light and point to specific empirical evidence and discussions, maybe?

- I think in a strategic and recursive reasoning scenario, it would be naive to assume fully cooperative agents or fully observable states. Human social behavior often includes non-verbal signals, contextual factors, and sometimes deceptive strategies. How does your model accommodate or account for these in multi-agent environments, particularly in scenarios with partial observability or intentional misinformation, such as a Byzantine Generals Problem (see [2] for instance)? This would address limitations of a strictly goal-oriented approach in highly dynamic social contexts.

[2] Konan et al. “Iterated reasoning with mutual information in cooperative and byzantine decentralized teaming”, ICLR 2022

- Related to the above question, your approach assumes that agents act in a goal-directed manner with underlying rational beliefs, is that correct? How robust is the model then when applied to agents with less predictable, impulsive, or irrational behaviors (again, see [2] for examples), which are common in social contexts? It would be valuable to understand the model's adaptability in scenarios where agent behavior deviates from optimal or rational planning.

**Strengths:**

-- See above

**Weaknesses:**

-- See above

**Questions:**

-- See above

---

> ### Author Response · Authors · 2024-11-27
> **Response to Reviewer KWEN**
>
> We thank the reviewer for great suggestions. We uploaded a revised pdf with changes highlighted in blue and addressed the reviewer's concerns point by point:
>
> **Choice of cognitive hierarchy.** We first define a forward process for different levels of cognition, generate a policy matrix based on the reward value map and chosen reasoning process, and then calculate the likelihood of observed state-actions pairs. The appropriate level of cognitive hierarchy was chosen by comparing the modeling fitness over the observed trajectories based on AIC or other potential model fitness criteria.
>
> **Computational cost.** Assume we have two agents executing independently in a task with $N$ states and $A$ actions, this expands to  $N^2$  states and  $A^2$ actions when considering joint state and action spaces. In our algorithm, we leverage PyTorch’s automatic differentiation to perform inference of the value function to maximize the likelihood of the observed state-action pairs. The computational complexity is equivalent to the complexity of the forward policy calculation process multiplied by the number of iterations required for convergence of the inference process, which is dependent on the specifics of the optimizer and the loss landscape. Our approach of value decomposition decreases the number of parameters to infer from $O(N^2)$ to $O(N)$, greatly accelerating the iteration process. But it’s hard to give an accurate estimate of the number of iterations needed due to complexity of the loss function. Therefore, we focus on the computational complexity of the forward policy calculation through value iteration.
>
> A level-0 agent follows a standard single-agent value iteration process, requiring the formation of a policy matrix of size $N \times A$. This process incurs a computational cost of  $O(K \times N^2 \times A)$ , where K is the number of iterations needed for the value iteration process. A level-1 agent first predicts that its counterpart is a level-0 agent, and then calculates its own policy based on this assumption, involving the formation of a joint optimal policy matrix of size  $N^2 \times A^2$. This increases the computational cost to  $O(K \times N^4 \times A^2)$ due to the complexity of value iteration at this level. A level-2 agent first forms a prediction model of a level-1 agent, which incurs a computational cost of  $O(K \times N^4 \times A^2)$. Subsequently, it computes its own policy through matrix multiplication, maintaining a computational complexity of  $O(2K \times N^4 \times A^2)$ overall. For level-$k$ ($k>2$) agent, the computational complexity is $O(k \times K \times N^4 \times A^2)$.  In summary, the entire computational complexity equals the cost for policy calculation times the number of iterations in the inference process. The cost of computing the policy increases exponentially with the number of agents and grows linearly with the cognitive level.
>
> **Trajectories availability.** Our algorithm is mainly devised for cognitive or neuroscience usage. The expert trajectories could come from recorded animal trajectories such as bat mimicking flying, fish schooling, rodent interactions. An existing open dataset is MaBe challenge https://www.aicrowd.com/challenges/multi-agent-behavior-challenge-2022.
>
> **Justify the recovered latent belief.** We believe the value maps recovered in Fig.2DE are valid because the actual reward location is lit up in these maps. In Fig.F, the value function of Euclidean distance also reflects the essence of a cooperative foraging task where animals need to be simultaneously at the same location to receive reward. We also plotted the occupancy maps (Fig. S7) of the observed trajectories ${(s_1^t, s_2^t, a_1^t, a_2^t)}$ as $a(s_1) = \sum_t \\delta(s_1^t)$, $b(s_2) = \sum_t \\delta(s_2^t)$ and $c(d) = \sum_t \delta (d(s_1^t,s_2^t))$. In the foraging case, $a,b$ looks like a more diffused version of recovered value map $m,n$ but the empirically observed interaction term $c$ has a peak when distance = 1 and the recovered value map highlights the joint states where distance=0. This is because exploration terminates at the rewarded state, causing a smaller occupancy frequency. But MAIRL still recovers distance=0 as the critical state.  In the hallway task case, the occupancy map (Fig. S10)  has very high values on the middle row but the recovered value map in Fig.4C only highlights the reward locations at the two ends.
>
> **Accommodate for partial observability.** We included this as a limitation.
>
> **Irrational agents** Yes. We assume the agents act in a goal-directed manner governed by underlying value functions. The decisions are always made in an optimal manner given a chosen value function.  If agents behave irrationally by doing random explorations, our algorithm will discover a value function contaminated by some random variables. For example, we performed MAIRL on random walk trajectories. The recovered value map and interaction map are totally random (Fig.S5)

---

### Official Review · Reviewer_w7R4 · 2024-11-04

**Soundness:** 2
**Presentation:** 2
**Contribution:** 2
**Rating:** 3
**Confidence:** 4

**Summary:**

This paper introduces a novel approach to understand social behavior by inferring latent beliefs and intentions of agents. MAIRL leverages probabilistic recursive modeling and joint value decomposition to model complex social interactions. The method is validated on simulated tasks and real-world datasets involving humans and animals, demonstrating its ability to uncover goal-directed value functions, interaction terms, and cognitive hierarchy levels.

**Strengths:**

1. The combination of probabilistic recursive modeling and joint value decomposition for inferring latent beliefs in social interactions is interesting.
2. The method is applied to diverse scenarios, including simulated tasks and real-world experiments, demonstrating its versatility and effectiveness.

**Weaknesses:**

The presentation, especially Figure 1, could be clarified. Figure 1 is confusing, specifically regarding the following points:
	•	Could you clarify the meaning of “map” in the figure and in the section?
	•	Are all the labels at the top of the figure meant to be “Agent 1,” but in different colors? If so, why use different colored blocks to represent the same meaning?

Some expressions in the formulation appear incorrect or unclear: Is the statement in line 225 incorrect, as $\hat{a}_2$ should represent an action instead of a policy?

This work includes human datasets in the hallway task, yet there is no mention of ethics approval or dataset accessibility. It’s unclear if the dataset is publicly available or was collected by the authors. An ethics statement would be beneficial for transparency.  It is better to provide following documents: the ethics committee that approved the study, the consent process for participants, and details on how other researchers can access the dataset.

While the use of MAIRL to enhance Theory of Mind (ToM) capability is intriguing, the core idea feels relatively straightforward and lacks significant novelty. The decomposition approach is basic, and the experiments are limited in complexity. Exploring a more innovative and meaningful decomposition method could improve the work. Additionally, testing in more complex experimental environments, such as the two-player Overcooked video game or real ToM test tasks (inspired by study in psychology) [1], would provide a more robust evaluation.

[1] Strachan, J.W.A., Albergo, D., Borghini, G. et al. Testing theory of mind in large language models and humans. Nat Hum Behav 8, 1285–1295 (2024). https://doi.org/10.1038/s41562-024-01882-z

**Questions:**

"theory of mind (ToM)." in second line of Introduction should be "theory of mind (ToM)".

**Details Of Ethics Concerns:**

The paper uses collected human dataset without any approval and ethic statment.

---

> ### Author Response · Authors · 2024-11-27
> **Response to reviewer w7R4**
>
> We thank the reviewer for great suggestions. We uploaded a revised pdf with changes highlighted in blue and addressed the reviewer's concerns point by point:
>
> **The presentation of Figure 1** The blue boxes belong to agent 2's policy generation process. Map means a value function which takes location or Euclidean distance as input and outputs a real number as the value of that location. To ensure precise terminology and better alignment with Eq. 2, we have updated Fig. 1 to more accurately reflect its relationship with the equation.
>
> **$\hat{a^2}$ represent an action instead of a policy** Yes. it should be action. Thanks for catching the typo.
>
> **This work includes human datasets in the hallway task, yet there is no mention of ethics approval or dataset accessibility ** Sorry for the missing information. This dataset was collected by Ho et al in 2016 and is publicly available. The ethics statement was included in the original publication. We revised the corresponding text to emphasize that we were analyzing an open dataset.
>
> **The decomposition approach is basic, and the experiments are limited in complexity** We respectfully argue that the decomposition method is far from trivial, as kindly highlighted by reviewers KWEN and 85mY. The decomposition approach not only enables efficient inference but also enhances the interpretability of value function estimation. Additionally, we provided a formal analysis of its validity in Session 3.2 and Supplementary Section A.1, which allows users to assess the approximation error based on the task setup and chosen interaction terms before executing the full inference algorithm. This analysis also demonstrated the effectiveness of the decomposition method in a three-agent cooperative foraging task. We appreciate the reviewer’s thoughtful suggestion regarding task selection and will certainly explore the Overcooked task in future work. Thank you for your valuable feedback.
>
> **"theory of mind (ToM)." the second line of Introduction should be "theory of mind (ToM)".** We’d appreciate it if the reviewer could clarify on the question. They are exactly the same phrases to us.

---

### Official Review · Reviewer_ywtE · 2024-11-04

**Soundness:** 2
**Presentation:** 2
**Contribution:** 2
**Rating:** 3
**Confidence:** 3

**Summary:**

This paper considers the two-agent IRL problem for reward inference. As a solution, the authors modify the non-stationary IRL approach by Ashwood et al. (2022) and propose an approximation of the goal map consisting of two marginal maps and an interaction map. Additionally, they consider different levels of cognitive hierarchy, i.e., how deeply the agents' actions are related recursively. The proposed method is validated on a simulated cooperative foraging task and further applied to human data on the hallway task and monkey data on the chicken task.

**Strengths:**

First, the problem considered in this paper, to do IRL for the multi-agent setting, is interesting and relevant. Due to the complexity, there is not much work in this direction and I appreciate that the authors venture into this direction.

As mentioned, applying IRL to multiple agents can become difficult because the joint state space, to which the rewards need to be assigned, grows exponentially in the number of agents. The approximation that the authors propose, to have two marginal maps and an interaction map, which are learned and combined, seems an interesting and clever idea.

Also, the problem of modeling and inferring the cognitive hierarchy is non-trivial and relevant. A further strength of the paper is, therefore, that they propose a model and experimentally try to infer the level.

Finally, the method was applied on real human and monkey data to prove the applicability of the method.

**Weaknesses:**

While the problem the authors tackle is relevant and interesting, the proposed solution is very incremental to the work of Ashwood et al. (2022). As the paper discusses, the joint state space of the MAIRL problem poses a difficulty to existing IRL methods, as the number of states scales exponentially in the number of agents. While this is, to my understanding, the main selling point of the paper, the solution the paper proposes is limited to two agents, and in their experiments only two-player experiments are considered.
The approximation of the goal maps still seems interesting and probably could be extended to more than two players, but the paper lacks an analysis of the limitations of this approximation.

Further, I found the abstract and introduction imprecise and kind of overstated concerning the objectives of the proposed method. I would have wished for a more detailed description, of what kind of environments can be considered, what properties can be inferred… Also in the following sections, I had the impression that clarity could be improved (see also my questions on this).

The considered hierarchical models of behavior are highly simplified and I am unsure how realistic they are. The first model (Eq. 243) only considers random actions of the other agent. For the second model (Eq. 246), one assumes that the other agent acts according to the first model. I also think that the description of these models could be improved, as it took me a while to understand them (and I am still unsure if I understood them correctly and how optimal policies are finally computed). I am also unsure how meaningful the results are to infer the model underlying behavior in the experiments with AIC.

I am also unsure about the results of the real human and monkey experiments. The hallway task seems to indicate the limits of the Euclidian interaction map, showing that for each experiment there is the need to manually design and test different kinds of interactions. In the monkey experiment, there are not even states and, therefore, the main motivation for the applicability of the approach, a large joint state space, is not fulfilled and the method therefore not needed.

The limitations are only very superficially discussed (limited discrimination between the two models and first-level reasoning). There are many assumptions and limitations of the method that could be discussed but are not, e.g. agent’s beliefs, knowledge of the position of other agents, number of other agents, and different interaction maps.

**Questions:**

In the abstract, line 31, you write “MAIRL offers a new framework for uncovering human or animal beliefs in social behavior”. Where do you infer the agent's beliefs?

In the introduction, line 47, you write “our novel approach promises to reveal the latent model of the world and the computation of value and mental states of multiple interacting agents in social behavior”. Where do you infer world models and mental states?

Line 67: “While MARL with recursive reasoning models social behavior using known value functions, real-world interactions among humans or animals often lack explicitly defined value functions.” What do you mean by this? Do you have a reference for this?

Line 94: “While fruitful, this approach has several drawbacks. First, there is a lack of consistent mapping between the inferred value function and the specific mental state of the agent such as goal…” Could you elaborate on this? Why is the inferred value function for a mental state not consistent?

Line 101: “Alternatively, based on MDPs, …” Are the previously considered approaches not based on MDPs?

Eq. 1: This equation looks odd to me. What about the dependence of a state on the past state and action?

Figure 1: Why does agent 1 have two individual policies, which are combined?

Line 170: “In the case of cooperative open arena foraging, we showed that the joint reward could be perfectly reconstructed” Where?

Line 159: Do you also consider in your algorithm $K$ different goal maps? On the other hand, if you don’t reuse this equation from Ashwood et al. (2022), how do you use $u$ as defined in Eq.2?

Is there a marginalization over $a_2$ missing in the equation of $P(a_1 | s)$ (line 243)?

Could you explain Figure 2F in more detail, as in the caption I found it confusing what is meant with “...other who is placed at the origin for plotting purpose and marked with an asterisk”?

Why is column distance the best for the interaction map for the hallway task? Intuitively, I would assume that collisions happen if both agents remain in the same row and therefore a row distance of 1 would be more critical than a column distance of 0?
How can you create and test different interaction measures for a new application?

Fig. 4C: Why do you estimate maps independently for success and failure trajectories? In a setting where you don’t know the value map you usually cannot tell whether it was a success or failure, but this is what you want to learn.

---

> ### Author Response · Authors · 2024-11-27
> **Response to Reviewer ywtE**
>
> We thank the reviewer for great suggestions. We uploaded a revised pdf with changes highlighted in blue and addressed the reviewer's concerns point by point:
>
> **Limitations of value approximation.** We did a mathematical analysis of the validity of value decomposition and numerically evaluated the approximation errors in the Session 3.2 and Supplementary Session A1
>
> **Abstract and introduction.** We updated them
>
> **The considered hierarchical models of behavior are highly simplified...** We provided modeling frameworks for level-0, level-1 agents with chance predictions and level-1 agents with perfect predictions. These forward modeling processes may be mathematically simple but definitely not trivial in terms of modeling human/animal behavior as human exhibited a cognitive reasoning level of approximately zero or one. When applied to multi-state, multi-action and multi-agent cases, even level-1 agents could induce complicated behaviors and equilibriums. Using AIC to compare models is a well-established approach in statistical modeling. We'd be happy to incorporate additional model comparison metrics if the reviewer could kindly suggest alternatives.
>
> **Limits of the Euclidian interaction map...** A good pick of the interaction term is the key to reveal an interpretable value estimation in social tasks. In most cases, it’s hard to interpret the full joint value function because its function domain is too large. In the cooperative foraging task, it is reasonable to use a Euclidean distance interaction term because the task requires the agents to occupy the same grid simultaneously. And indeed, value estimated expert trajectories highlight a huge value at distance equals to zero. In the hallway task, the goal is to switch locations with fewer steps. It is not immediately clear to us which interaction term could be most essential. Therefore, we tested four different interaction terms and concluded that column distance is the most essential one. This reveals the essence of the task: switching column indices.  The monkey experiment is to showcase the applicability of our method to a non-cooperative task.
>
> **Limitations.** We rewrote the paragraph.
>
> **Where do you infer the agent's beliefs?** We intend to infer the belief from expert trajectories, usually collected through cognitive or neuroscience experiments. An example dataset: https://www.aicrowd.com/challenges/multi-agent-behavior-challenge-2022.
>
> **Where do you infer world models and mental states?**  We removed this phrase.
>
> **Line 67...** We revised the sentence as “While MARL with recursive reasoning typically models social behavior in goal-directed tasks, ...”  In real-world interactions, explicit value functions are often unavailable or poorly defined, unlike in constrained task or game setups. Naturalistic behaviors typically emerge from complex, latent processes involving subjective preferences, hidden goals, and environmental factors. For instance, animal behaviors such as foraging or group movement frequently involve implicit trade-offs (e.g., balancing safety and food acquisition) that are not explicitly represented as value functions.
>
> **Line 94...** The sentence in question was from the review paper by Jara-Ettinger (2019, Fig. 1 legend). However, we recognize that mentioning it here without sufficient context could lead to confusion and removed it.
>
> **Line 101** We removed it.
>
> **Eq. 1** We assumed state-action pairs are independent along the temporal sequence, a common assumption used in most IRL literature.
>
> **Figure 1**  The blue boxes belong to agent 2's policy generation process. We updated Figure 1.
>
> **Line 170** Please refer the Supplementary Session A1 and Figure S1, S2 and S3.
>
> **Eq2** We assumed K=1. To avoid confusion, we removed the summation over $k$. (Eq.2)  Line 243: We have plugged in the marginalization expressions ($P(\hat{a_2}|s)$) based on random explorations.
>
> **Figure 2F** We replotted Fig.2F with the horizontal axis as the Euclidean distance between agents.
>
> **Column distance in the hallway task** In the hallway task, two agents cannot occupy the same grid. Agent 1 (on the left) tends to move to the right, while Agent 2 tends to move to the left (Fig.S10). A state where the column distance is zero indicates that the two agents are on different rows and successfully passing each other. However, a row distance of one can also occur when the agents are still far apart column-wise, which does not necessarily ensure successful task completion. Choosing the interaction map depends on the specific task. However, we could validate whether the decomposition is valid or tolerable using the analysis in Supplementary Session A1.
>
> **Fig. 4C** The trial types were predetermined by the rule. Failure trial: only one participant arrives at the goal. Success trials: both participants get to the goal simultaneously. Expert trials: both participants get to the goal simultaneously in minimum steps (7 steps).

---

### Author Response · Authors · 2024-11-27
**Global Response to all reviewers**

We'd like to thank reviewers for constructive comments. We have incorporated most changes in our revised manuscript (highlighted in blue). The major changes that are requested by most reviewers are:
- We rewrote the 'related work' and 'limitation' session.  In 'related work', we reviewed theoretical advances and common assumptions for MAIR in imitation and efforts in cognitive sciences to model social interactions based on single-agent Partially Observable Markov Decision Process (POMDP) or multi-agent Markov Decision Process (MDP).
- We performed a rigorous mathematical analysis and numerical evaluation of the validity of value decomposition in Session 3.2 and Supplementary Session A1. We numerically evaluated the reconstruction error in the two-agent foraging and hallway task and found them to be three or four orders of magnitude smaller than the actual reward strength (Fig.S2). When reward location is not sparse, the interaction term becomes essential to maintain a low reconstruction error (Fig. S3). More interestingly, when three agents are considered
in the task, we could still reconstruct the joint value function using their individual maps and two pairwise interaction terms with reconstruction error four orders of magnitude smaller (Fig. S4). These mathematical analysis results justified the validity of value decomposition and indicate the potential of our method to be applied to tasks with more than two agents.

---

### Meta-Review · Area_Chair_vywy · 2024-12-20

**Metareview:**

This paper proposed a MARL-based model of social behavior in humans and animals. Unfortunately, reviewers had concerns about validity, evaluation methodology, and appropriate referencing of related work.

**Additional Comments On Reviewer Discussion:**

All reviewers indicated that they read the authors' response and some changed their scores.

---

### Decision · Program_Chairs · 2025-01-22

Reject